# Activation of Transposable Elements in Immune Cells of Fibromyalgia Patients

**DOI:** 10.3390/ijms21041366

**Published:** 2020-02-18

**Authors:** Tamara Ovejero, Océane Sadones, Teresa Sánchez-Fito, Eloy Almenar-Pérez, José Andrés Espejo, Eva Martín-Martínez, Lubov Nathanson, Elisa Oltra

**Affiliations:** 1School of Medicine, Universidad Católica de Valencia San Vicente Mártir, 46001 Valencia, Spain; tamara.ovejero@ucv.es; 2Université de Poitiers, CEDEX, 86073 Poitiers, France; sadonesoceane@gmail.com; 3Escuela de Doctorado, Universidad Católica de Valencia San Vicente Mártir, 46008 Valencia, Spain; mt.sanchez@ucv.es (T.S.-F.); eloy.almenar@ucv.es (E.A.-P.); 4School of Biotechnology, Universidad Católica de Valencia San Vicente Mártir, 46001 Valencia, Spain; joseandres.espejo@mail.ucv.es; 5National Health Service, Manises Hospital, 46940 Valencia, Spain; evamariamartinmartinez@gmail.com; 6Institute for Neuro Immune Medicine, Nova Southeastern University, Ft Lauderdale, FL 33314, USA; lnathanson@nova.edu; 7Centro de Investigación Traslacional San Alberto Magno, Universidad Católica de Valencia San Vicente Mártir, 46001 Valencia, Spain

**Keywords:** fibromyalgia, myalgic encephalomyelitis/chronic fatigue syndrome (ME/CFS), human endogenous retrovirus (HERV), transposable elements, epigenetics, DNA methylation, transfer RNA small fragments (tsRNAs), interferon (INF), non-Hodgkin’s lymphoma

## Abstract

Advancements in nucleic acid sequencing technology combined with an unprecedented availability of metadata have revealed that 45% of the human genome constituted by transposable elements (TEs) is not only transcriptionally active but also physiologically necessary. Dysregulation of TEs, including human retroviral endogenous sequences (HERVs) has been shown to associate with several neurologic and autoimmune diseases, including Myalgic Encephalomyelitis/Chronic Fatigue Syndrome (ME/CFS). However, no study has yet addressed whether abnormal expression of these sequences correlates with fibromyalgia (FM), a disease frequently comorbid with ME/CFS. The work presented here shows, for the first time, that, in fact, HERVs of the H, K and W types are overexpressed in immune cells of FM patients with or without comorbid ME/CFS. Patients with increased HERV expression (N = 14) presented increased levels of interferon (INF-β and INF-γ) but unchanged levels of TNF-α. The findings reported in this study could explain the flu-like symptoms FM patients present with in clinical practice, in the absence of concomitant infections. Future work aimed at identifying specific genomic loci differentially affected in FM and/or ME/CFS is warranted.

## 1. Introduction

Fibromyalgia (FM) (ICD-10 diagnosis code M79.7) [1] is defined as a chronic widespread pain disorder of unknown etiology persisting for more than three months in the absence of any obvious organic lesion. Symptoms commonly associated with FM include low pain threshold, joint stiffness, sleep disturbance, cognitive dysfunction, fatigue and depression [2,3,4]. In addition, mechanisms tied to FM symptomatology include changes in nociceptive circuitry and an increase in pain sensitivity [5].

FM frequently presents with comorbid Myalgic Encephalomyelitis/Chronic Fatigue Syndrome (ME/CFS; ICD-10 code R53.82 or G93.3 if post-viral). Although separate clinical diagnostic criteria are described for FM and ME/CFS [2,3,6,7,8,9,10], both conditions are associated with common features including alterations in immune function and overlapping infectious symptomatology such as flu-like symptoms [11,12]. As a result, researchers have investigated a possible viral etiology of the disease [13,14]. However, to date, no clear correlation between FM or ME/CFS and viral infection has been established.

A higher prevalence in females has been reported for both FM and ME/CFS. Global estimates range from 1% to 5% of the population for FM and from 0.23% to 5% for ME/CFS depending on geographic areas and the diagnostic criteria applied [15,16].

Most chronic and degenerative disorders seem to derive from complex gene-environment interactions leading to aberrant epigenetic changes rather than being determined by genetic mutations or polymorphisms. A recent report by Polli et al., summarizes the available evidence connecting epigenetic mechanisms to pain [17], particularly on DNA methylation changes and miRNA interference. DNA methylation profiles and the histone posttranslational modification (PTM) landscape shape mammalian genomes, dynamically controlling gene expression by altering chromatin organization [18], while miRNA profiles (miRNomes) post-transcriptionally regulate target RNA levels [19].

So far, only two studies have tried to evaluate genome-wide DNA methylation profiles in patients with FM [20,21] finding differential methylated regions associated to 47 and 960 genes respectively, with some overlaps. The main affected genes relate to DNA repair, immune system, nervous system and skeletal/organ system development, and chromatin compaction pathways [17,22]. With regards to miRNA profiling, five studies have reported FM miRNomes [23,24,25,26,27], including research efforts from our group. 

Like DNA methylation studies, miRNA screenings typically have included a low number of participants but have still resulted in overlapping deregulated miRNAs identified by more than one group of researchers. For a detailed recompilation of up-to-date results, a recent review published by our group is suggested [28].

Interestingly, as described for FM, ME/CFS patients have shown predominant hypomethylated DNA patterns within differentially methylated regions [29,30,31]. Genome hypomethylation can lead to transcriptional activation of regions of the genome that would appear otherwise silenced. Although coding genes only constitute about 2% of the genome, they have traditionally taken most of the attention in many differential expression studies. However, after noticing that many of the hypomethylated regions in ME/CFS lie within non-coding regions [30], we aimed to determine whether these epigenetic marks in ME/CFS could affect the expression of repetitive regions, representing 45% of the genome [32]. This has been shown for other neurological and autoimmune diseases, such as Multiple Sclerosis (MS), Amyotrophic Lateral Sclerosis (ALS), Rett Syndrome and Systemic Lupus Erythematosus (SLE) [33].

As previously published by our group, it seems possible that the activation of particular patterns of transposable elements (TEs) associating with ME/CFS epigenetic marks may trigger ME/CFS patient flu-like symptoms in the absence of concomitant active infections [34]. Further work after this publication by Dr. Romano′s group at the Universidade de Sao Paulo in Brazil, in fact supports that the levels of the endogenous retrovirus HERV-K are upregulated in ME/CFS. This has been demonstrated by an analysis of peripheral blood mononuclear cells (PBMCs) isolated from ME/CFS patients at the monographic United Kingdom (UK) ME biobank [35], supporting the hypothesis raised by our group that TE activation may contribute to ME/CFS [34]. However, and although FM patients frequently present comorbid ME/CFS syndrome, no study has yet determined whether patients with a primary diagnosis of FM presents a similar activation of this group of TEs. In addition, no study has evaluated the molecular consequences of TE activation in FM or ME/CFS.

The aim of the present study was to explore whether HERV elements, constituting an 8% of the human genome, are also activated in FM, with and without comorbid ME/CFS, and in addition determine whether the increased levels of HERV transcripts (if present) affects patients′ immune system physiology.

## 2. Results

### 2.1. Demographics and Other Characteristics of Participating Individuals

Average age of participating patients was 54 ± 7.4 years (range 42–65) and 50.4 ± 10.1 (range 38–65) for the matched healthy control (HC) group. At least 80% of FM patients (11/14) suffered the disease by more than 3 years and over 40% (6/14) by over 12 years. All participants were female (N= 28; 14 FM and 14 HCs), and all individuals in the patient group fulfilled the revised American College of Rheumatology (ACR) criteria for the diagnosis of FM, as described in the Methods section [2,3]. In addition, 50% (7/14) fulfilled the Canadian criteria for ME/CFS and all but one of the seven participants presenting comorbid ME/CFS also fulfilled the International criteria for the disease [7,8]. By comparing and contrasting molecular patterns of FM patients presenting or not comorbid ME/CFS disease-specific changes may be observed.

Total Fibromyalgia Impact Questionnaire (FIQ) average score of FM participants was 74.88 ± 12.59 (range 56.30–92.93) and Multi Fatigue Inventory (MFI)′s for general fatigue was 17.31 ± 3.23 (range 10–20). Summary FIQ, MFI and quality of life SF-36 questionnaire scores for participating patients (N = 12–13) are shown on Table 1. Complete itemized questionnaire scores are shown on Appendix A.

A detailed list of drugs and supplements regularly taken by participating patients during at least the 6 months preceding study enrolment is provided on Appendix A. Main group prescriptions include analgesics/anti-inflammatories, antacids, antilipemics, anticonvulsants, antidepressants, antihistamines, antihypertensives, anxiolytics and other. Most participants (over 75%) also used dietary supplements. Participating patients withdrew medication 12 h prior to blood, as described in Methods. 

### 2.2. Overexpression of HERV Sequences in FM

Based on the hypothesis that the hypomethylation patterns detected in ME/CFS and the altered miRNA levels in both FM and ME/CFS may associate with aberrant activation of TEs [34], we evaluated whether participating patients present with increased levels of HERVs, a subtype of TEs that can mimic infection. 

Although retrotranscription followed by quantitative PCR (RT-qPCR) amplification of HERVs does not provide location specific information of the amplified sequences, we decided to take this overall estimation approach using sets of primers formerly described by Johnston et al., [40].

As shown in Figure 1, the FM patients participating in this study presented with increased levels of HERV-H, HERV-K and HERV-W in their PBMCs, with respect to the levels registered in that same blood fraction of HCs.

### 2.3. Interferon and TNF-α Levels in PBMCs of FM Patients Showing HERV Activation

In an effort to further understand the effects of transcriptional activation of HERVs on the FM immune system and reported inflammation of FM participants [4,11,41], PBMC cellular interferon and tumor necrosis alpha (TNF-α) levels were measured. Figure 2 shows that the patients with increased expression of HERVs, also presented higher levels of interferon (INF-β and INF-γ) as determined by RT-qPCR quantitation. In fact, positive correlations between HERV-H and INF- β levels, in addition to significant correlations between HERV-W with both, INF β and INF-γ, were found (Figure 3). Levels of TNF-α mRNA, however, were not significantly affected although values presented higher variability across FM participants than in the HC group (Figure 2).

Positive significant correlations were also detected between HERV-H and HERV-W levels (Figure 4). Other combinations tested did not show statistical significance (Figure 3 and Figure 4).

Although induction of interferon production was expected in response to increased dsRNA levels deriving from activation of HERVs, the unchanged levels of TNF-α observed were somehow unexpected as inflammation often associates with increased TNF-α levels [42].

### 2.4. tRNA Levels in FM Patients Overexpressing HERVs

Interferon production activates endonucleases as part of the cellular response mechanisms to degrade invading virus, along with INF-stimulated genes (ISGs) [43,44,45]. 

Donovan et al., recently showed that activation of the RNase L enzyme, a downstream target of INF signaling, leads to fragmentation of tRNAs (tsRNA or transfer RNA small fragments) for Histidine (His) and Proline (Pro), even before protein synthesis is shut down [46]. With the idea that these tsRNAs could therefore constitute surrogate markers of the activation status of RNase L, we examined whether patient PBMCs presented differences in the content of these tRNAs with respect to HC′s. 

As shown in Figure 5, reduced levels of tRNA-His and tRNA-Pro were detected in PBMCs of some of the FM patients presenting increased HERV and INF levels with respect to HC′s. Although differences found were not statistically significant, tRNA levels showed a reduction tendency (*p* < 0.1), at least for tRNA-Pro contents. Significant correlations between tRNAs, HERV and INF levels were not found either (data not shown). In addition, the assay did not allow detection of tsRNAs.

## 3. Discussion

The importance of the present study relies in that it shows, for the first time, that the activation of TEs, in particular some HERV sequences, is a mechanism linked to FM, potentially explaining the reasons for repeated failures in detecting exogenous infectious agents as etiologic triggers of the disease and for the flu-like symptoms patients experience [11,12,13,14]. However, reduced sample size and inability to identify specific activated genomic loci with the method used (limitation of approach), prevented detection of molecular TE patterns between FM patients with or without comorbid ME/CFS. Still, the results obtained by Rodrigues et al., [35] demonstrated activation of HERV-K elements only and no changes in HERV-W levels, potentially supporting differential TE activation across ME/CFS patient cohorts. It remains unclear whether the participating patients in Rodrigues study presented with or without comorbid FM. 

Although a genetic link associated to FM cannot be ruled out at present, evidence supporting a relevant role of environmental factors in FM and ME/CFS is growing [28,30,47,48]. It should be mentioned at this point that some single nucleotide polymorphisms (SNPs) have been correlated with FM and ME/CFS [22,49] which, does not exclude the participation of epigenetic mechanisms in the pathophysiology of the disease.

Interestingly, the infection mimicry state possibly derived from the presence of complementary RNAs originated from transcription from either HERV end (HERV activation) in the studied subjects, correlates with higher INF-β, for both HERV-H and HERV-W HERVs, and increased INF-γ levels in patients positively correlates with HERV-W, in the absence of the inflammatory marker TNF-α (Figure 1, Figure 2, Figure 3 and Figure 4).

In relation with the findings of this study, the higher prevalence of musculoskeletal manifestations found in patients with malignant disease [50] and the increased prevalence of non-Hodgkin lymphoma in ME/CFS [51], we propose a model by which environmental factor mediated de-repression of TEs trigger INF response and cleavage of tRNAs. Patients with compromised RNase activity failing to degrade tRNAs could use intact tRNAs to retrotranscribe HERV RNA sequences, leading to genome instability. By contrast, the activated “copy and paste” TEs would not be retrotranscribed due to priming failure by fragmented tRNAs (tsRNAs) [52], preserving genome integrity under a normal wild type RNase L activity scenario (Figure 6).

Therefore, it may be hypothesized that an increase in long dsRNA intracellular levels in FM and ME/CFS patients deriving from epigenetic changes, as described [20,21,22,23,24,25,26,27,28,29,30,31,34], could in fact constitute a trigger for the disease in the absence of viral or other type of infection. Rodrigues et al., [35] and the results of this study provides evidence to support this hypothesis. It is perhaps the particular type of activated elements what drives the individual phenotype towards autoimmunity (i.e., through the synthesis of antigenic peptides encoded by TEs), neurological, or other health defects [33]. The fact that HERV-W immunopathogenic envelope protein (*Env*) associates with MS [53] and that HERV-E transcriptional activation in CD4+ T cells correlates with SLE [54] further supports our proposed model (Figure 6).

The approach used in this study uses degenerate primers for RT-qPCR to amplify a group of elements and therefore is limited at providing information on the specific locations of the genome contributing to increased HERV levels. The recent report by Rodrigues et al., [35] used different sets of primers and sets of samples, and yet, differential TE expression was detected, supporting the robustness of this unfocussed overall approach. Still, to identify FM and ME/CFS-specific activated TEs, future studies are suggested that use more specific approaches, such as microarray-based HERV subtyping [55] or RNA-Seq, after differential methylation DIP (DNA immunoprecipitation), followed by repetitive sequence pipeline analysis [56]. Another approach developed by Dr Mallet’s group at bioMériux, France, HERV-V2 chip, may be useful in identifying commonly activated HERVs. This approach uses an array dedicated to a collection of 5573 HERVs constructed with 23,583 probe sets (88,592 probes) to assign unique genomic position to these many sequences [55]. In addition, the use of TEtranscripts package [56] and the Software for Quantifying Interspersed Repeat Expression (SQuIRE) [57] are viable options to evaluate differential expression of TEs from RNA-Seq results.

As proposed by Donovan et al., [46] PBMC intracellular pools of tRNA-His and tRNA-Pro, may constitute surrogate markers of RNase L activity and thus, of TE activation. Although this possibility needs to be validated by complementary analysis of RNase L activity, the possibility of using this quantitative simple approach seems attractive. The northern blot method used in this study could be improved by the RtcB RT-qPCR method. The approach relies on the use of the RtcB ligase joining single stranded RNA with a 3’-phosphate or 2’,3’-cyclic phosphate to another RNA with a 5’-hydroxyl group prior to RT-qPCR amplification or RNAseq of RNase L cleavage products [46,58]. This alternative approach should elevate the sensitivity of the assay used in our analysis (northern blot, Figure 5) for a low requirement of total RNA towards quantitating tsRNA fragments.

Although the levels of TNF-α have been reported increased in the blood of FM and ME/CFS patients by some researchers, its association with disease remains controversial [41,59]. It should be highlighted that TNF-α levels are commonly assayed in serum while we used total RNA from PBMCs in our assays. Although TNF-α can be synthesized and released by some PBMC subpopulations such as CD4+ T lymphocytes, and natural killer (NK) cells; macrophages and mast cells are the main producers or stores of this cytokine [60]. In this sense it is worth mentioning that Olsnes et al., noticed that following *S. pyogenes* bacterial stimulation only monocytes in transit to become macrophages secrete TNF-α, but not by peripheral blood monocytes [61]. Perhaps analysis of IL-10 levels may help clarify the potential involvement of compensatory anti-inflammatory pathways along with the inflammatory cascade in FM in future HERV studies. Interestingly, it has been reported that IL-10 reduces pain perception by decreasing the level of IL-6 and TNF-α production by monocytes [62]. 

Moreover, tsRNA fragments, as well as other small RNA products of RNase L activity may interfere with the miRNA processing machinery, by their capacity of binding the Drosha and Argonaut (AGO) components of the microRNA biogenesis pathway [63]. This may explain, at least in part, the reported differential miRNA patterns (miRNomes) in FM and ME/CFs [23,24,25,26,27,28] and in additional neuroimmune diseases where transcriptional activation of HERVs has been reported [64]. Therefore, miRNomes, may be also informative of the activated state of TEs. Particular miRNomes preferentially associating to RNase L activity, TE activity and disease status remain to be identified.

Lastly, it should be highlighted that the participants of this study enrolled only after a minimum 12 h period of medication withdrawal prior to blood draw (please see the Methods section for details). As documented in our recent publication [28] and recommended by the NINDS ME/CFS Common Data Elements initiative [65], molecular biomarker research of FM and ME/CFS should restrict participation when possible or count with careful medication registry to minimize and control drug-associated biases. Associations between HERV and INF expression and medication intake remain presently unknown. 

## 4. Materials and Methods 

### 4.1. Participating Individuals and Associated Data

This study was approved by the Public Health Research Ethics Committee DGSP-CSISP of Valencia, núm.20190301/12, Valencia, Spain. Patients were invited to participate by advertising the study at local patient associations. HCs were invited through the Umivale mutual health insurance company, Valencia, Spain, during their routine annual checkup visit, to avoid additional phlebotomies. Written informed consent was obtained from all study participants. HCs were matched by age (±5 years) to participating patients.

Patients, all female, underwent a thorough clinical interview and medical examination to assess clinical criteria for FM, using the 2011 American College of Rheumatology (ACR) criteria [2,3] and ME/CFS comorbidity according to Canadian [7] and/or International Consensus [8] criteria. Patients with health problems other than FM and ME/CFS were excluded. Patient health status was also evaluated with the use of standardized questionnaires, including the FIQ case report form [36,37], the MFI questionnaire [38], and the quality of life SF-36 instrument [39]. Participating patients agreed to withdraw medication at least 12 h prior to blood draw. Participating HCs were included only if not having a medical history of chronic pain and/or fatigue, or serious health complications. Medicated HCs were also excluded. A single 10–20 mls sample was provided per participant.

### 4.2. PBMC Isolation and Total RNA Extractions

For the isolation of PBMCs, 10 mls of blood were collected in K2EDTA tubes (Becton Dickinson, Franklin Lakes, NJ, USA) and processed within 2 h by dilution at 1:1 (v/v) ratio in phosphate-buffered saline solution (PBS), layering over one volume of Ficoll-Paque Premium (GE Healthcare, Chicago, IL, USA) and separation by density centrifugation at 500× *g* for 30 min (20 °C, brakes off). The PBMC layer was washed with PBS and resuspended in red blood cell lysis buffer (155 mM NH4Cl, 10 mM NaHCO3, 0.1 mM EDTA, and pH 7.4), kept on ice for 5 min, and centrifuged (20 °C at 500× *g* for 10 min), to remove contaminating erythrocytes. The washed pellets were adjusted to a final concentration of 10^7^ cells/mL in freezing medium (90 % FBS, 10 % DMSO), aliquoted and deeply frozen in liquid nitrogen until use. Total RNA was extracted with RNAzol (Molecular Research Center, Cincinnati, OH, USA) according to the manufacturer’s instructions. RNA quality was assessed using Agilent TapeStation 4200 (Agilent Technologies, Santa Clara, CA, USA). Only RNA samples with RNA Integrity numbers (RIN) above 7 were further analyzed.

### 4.3. RT-qPCR Amplification

Reverse-transcription was performed with the High-Capacity cDNA reverse Transcription kit (Applied Biosystems, Waltham, MA, USA, cat. 4308228), using 1µg of total RNA according to manufacturer’s guidelines. cDNAs were used for Real time PCR using the PowerUP Sybr Green Master Mix (Applied Biosystems, cat. 100029283) and a Lightcycler LC480 instrument (Roche, Penzberg, Germany). Standard amplification conditions were used, including a single hotstart polymerase preactivation cycle at 94 °C for 15 min, and up to 45 amplification cycles, each one consisting of three steps: denaturation at 95 °C for 15 s, annealing at 50–60 °C for 30 s and extension at 70 °C for 30 s. Sequences of specific primers used are detailed in Table 2. GAPDH levels were used for the relative quantification of the RNAs amplified, 2^−ΔΔ*C*t^ analysis to calculate fold-change was applied.

### 4.4. Small RNA Northern Blot Analysis

In total, 3 µg of total RNA were separated in 15% denaturant polyacrylamide gels (urea 7M) and run 1 h at 300 V, as previously described [69]; transferred to Hybond-N+ nylon membrane (GE Healthcare, USA) with transfer buffer (trisodium citrate 6mM, Sodium phosphate dibasic 8mM) for 2 h at 350 mA at 4 °C. Then, membranes were cross-linked with ultraviolet light (UV) at 1200 µjoules during 1 min and hybridized with 5’biotin-labeled probes (Integrated DNA technologies, Leuven, Belgium) specific to tRNA-His (5’-CAG AGT ACT AAC CAC TAT ACG ATC ACG GCC-3’), to tRNA-Pro (5’- CCG AGA ATC ATA CCC CTA GAC CAA CGA GCC-3’) or to RNU6 (5’- CGA ATT TGC GTG TCA TCC TTG-3’) for normalization, as described by Donovan et al., [46]. Hybridization proceeded after membrane blocking, with 50–100 pmol/mL overnight at 40 °C with shaking, in hybridization oven. After three washes (10–15 min at RT with with washing buffer 1× SSC, 0.1% SDS), the membrane was developed with streptavidin-horseradish peroxidase (HRP) conjugate (ThermoFisher Scientific, Waltham, MA), at final concentration of 125 pg/mL in pre-hybridization buffer and ECL™ Prime Western Blotting System in an ImageQuant LAS 4000 Mini (GE Healthcare). Signals obtained were quantified with the Image J software (Bethesda, DC, USA) [70]. 

### 4.5. Statistical Analysis and Plotting

Continuous data are expressed as means ± SD, as indicated. Statistical differences were determined using two-tailed unpaired *t*-tests. Differences between groups were considered significant when *p* < 0.05. Analysis were conducted with the SPSS package 13.0 (SPSS Inc, Chicago, IL, USA). Variable correlations were evaluated by the simple linear regression method (least-squares approach). Plots were drawn using the GraphPad Prism 5.0 program (San Diego, CA, USA).

## 5. Conclusions

To the best of our knowledge this is the first study to report increased expression of HERV-K, HERV-H, HERV-W and INF-β and INF-γ levels correlations, in the immune system of FM patients.

Although the levels of these molecules may serve as biomarkers of FM and/or ME/CFS and/or biosensors of TE activation, the RT-qPCR overall estimation approach used in this pilot study may turn unspecific, broadly associating TE activity with neurological and inflammatory processes. Therefore, future efforts evaluating activation of particular HERVs or TE chromosomal sites should more precisely define disease-specific mechanistic information. 

Importantly, the model proposed here linking disease-specific epigenome modifications, TE activation, inflammation and an increased risk of cancer in individuals with compromised RNase activity (Figure 6) might be applicable not only to FM and ME/CFS patients but in any individuals with similar molecular disorders.

## Figures and Tables

**Figure 1 ijms-21-01366-f001:**
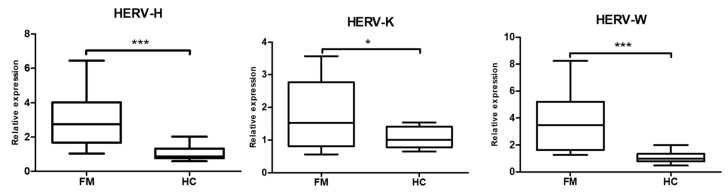
HERVs are overexpressed in FM. RT-qPCR amplification of HERV-H, HERV-K and HERV-W using total RNA from PBMCs is shown (N = 14/group) (*** *p* < 0.001; **p* < 0.05). Primer sets are detailed in Table 2 and conditions used described in Methods. Relative expression levels were calculated as 2^−ΔΔ*C*t^ values using GAPDH levels as reference. Group means and SEM values are shown.

**Figure 2 ijms-21-01366-f002:**
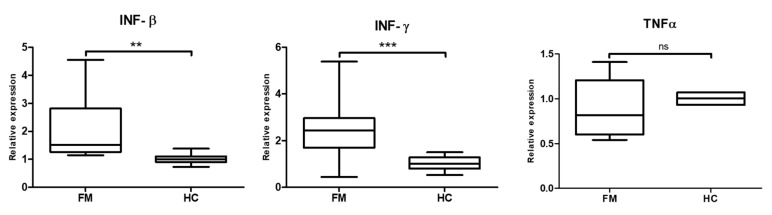
INF-β and INF-γ levels are increased and TNF-α are unaffected in FM patients showing increased HERV levels. RT-qPCR amplification of INF-β and INF-γ and TNF-α are shown (N = 14/group); (*** *p* < 0.001; ** *p* < 0.005); ns (non-significant). Primers sets used are detailed in Table 2 and conditions described in Methods. Relative expression was calculated as 2^−ΔΔ*C*t^ values using GAPDH levels as reference. Group means and SEM values are shown.

**Figure 3 ijms-21-01366-f003:**
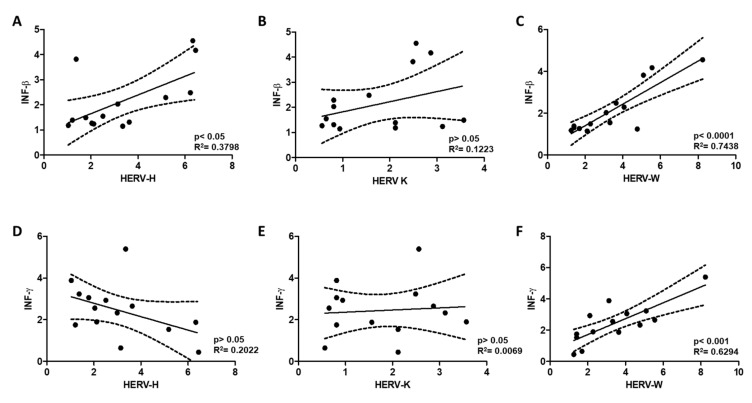
Linear regression analysis of HERV-H, HERV-K and HERV-W levels with respect to INF-β (**A**–**C**) and INF-γ (**D**–**F**) levels in PBMCs of FM patients. Adjusted calculated best-fit lines together with SD values (dotted lines) are represented for each data set pair; R^2^ and *p* values are indicated.

**Figure 4 ijms-21-01366-f004:**
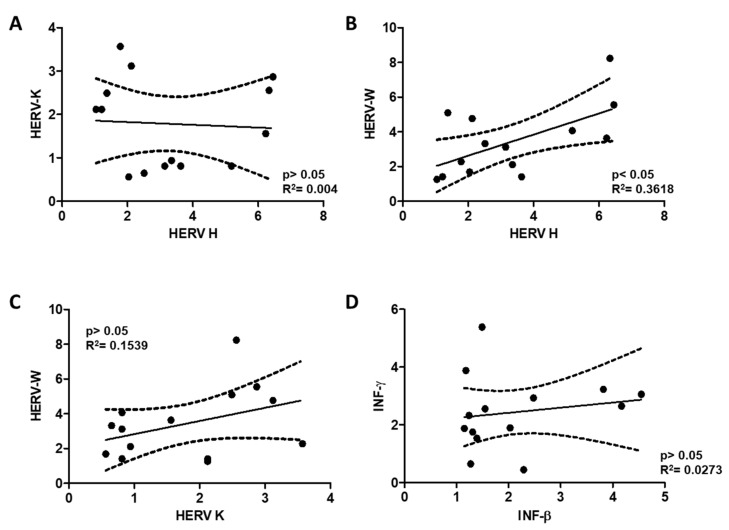
Linear regression analysis of HERV-H, HERV-K and HERV-W levels with respect to each other (**A**–**C**); and correlation between INF-β and INF-γ levels (**D**) in PBMCs of FM patients are shown. Adjusted calculated best-fit lines together with SD values (dotted lines) are represented for each data set pair; R^2^ and *p* values are indicated.

**Figure 5 ijms-21-01366-f005:**
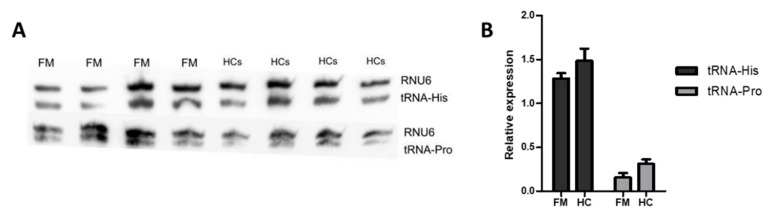
Northern blot analysis of tRNA levels in PBMCs from FM or HC participants, as indicated (**A**). Quantitated tRNA-His, tRNA-Pro levels (Image J software) upon normalization to RNU6 levels are shown (**B**) (N = 4/group).

**Figure 6 ijms-21-01366-f006:**
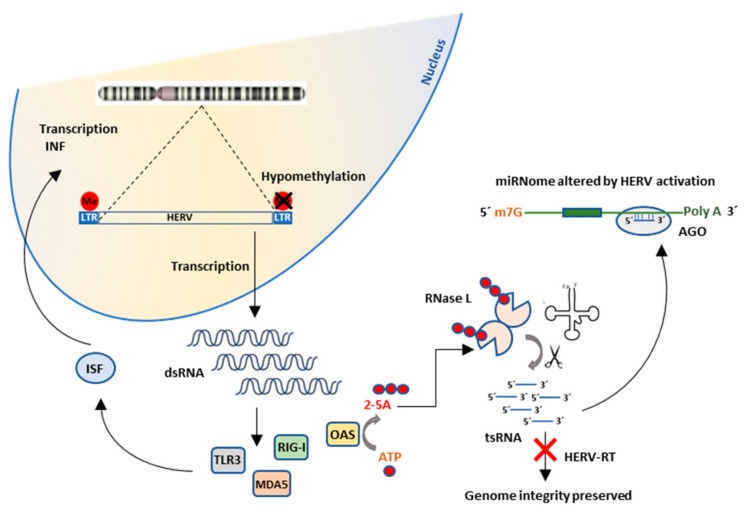
Proposed mechanism linking overexpression of HERVs, INF induction and increased cancer susceptibility in patients with defective ribonuclease activity. ISF (Interferon stimulating factors); TLR3 (Toll-like receptor 3); MDA5 (Melanoma differentiation associated gene 5); RIG-I (Retinoic acid-inducible gene I); OAS (2’-5’-oligoadenylate synthetase); 2-5A (2’, 5’ oligomers of ATP); tsRNAs (transfer RNA small fragments); RT (reverse transcription); AGO (Argonaut).

**Table 1 ijms-21-01366-t001:** Patient assessment with FIQ, MFI and SF-36 [36,37,38,39] questionnaires (N = 12–13).

Questionnaire	Mean	SD±SE	Range
**FIQ**			
Total FIQ	74.88	12.59±3.49	56.30-92.93
Function	5.76	1.847 ± 0.51	2.64-8.91
Overall	7.81	3.58 ± 0.99	1.43-10.01
Symptoms	6.38	2.955 ± 0.82	4.29-10.01
**MFI**			
General Fatigue	17.31	3.23 ± 0.89	10-20
Physical Fatigue	17.00	2.97 ± 0.82	12-20
Reduced Activity	17.00	2.97 ± 0.82	12-20
Reduced Motivation	15.62	3.07 ± 0.85	11-20
Mental Fatigue	16.15	2.97 ± 0.82	12-20
**SF-36**			
Physical Functioning (PF)	33.75	17.73 ± 5.12	5-65
Role Physical (RP)	0.00	0.00 ± 0.00	0
Bodily Pain (BP)	22.08	17.48 ± 5.05	0-57.5
General Health (GH)	18.33	17.49 ± 5.05	0-45
Vitality (VT)	12.92	11.37 ± 3.28	0-35
Social Functioning (SF)	28.54	20.27 ± 5.85	0-77.5
Role Emotional (RE)	25.00	45.23±13.06	0-100
Mental Health (MH)	47.67	19.48 ± 5.62	28-80

FIQ (Fibromyalgia Impact Questionnaire), MFI (Multi Fatigue Inventory) and SF-36 quality of life questionnaire. SD (standard deviation); SE (standard error). Range refers to the possible values in the studied group.

**Table 2 ijms-21-01366-t002:** Sequences of primers used in qPCR amplifications. Nucleotide ambiguities are labeled in red (Y = C or T; R = A or G).

Primer	Sequence	Reference
HERV-W F	5′-GGCCAGGCATCAGCCCAAGACTTG-3′	[40]
HERV-W R	5′-CTTTAGGGCCTGGAAAGCCACT-3′	
HERV-H F	5′-CTTTTATTACCCAATCTGCTCCCGAYAT-3′	[40]
HERV-H R	5′-TTTAGTGGTGGACAGTCTCTTTTCCARTG-3′	
Interferon-β-F	5′-ACCTCCGAAACTGAAGATCTCCTA-3′	[66]
Interferon-β-R	5′-TGCTGGTTGAAGAATGCTTGA-3′	
Interferon-γ-F	5′-GTGGAGACCATCAAGGAAGACA-3′	self-designed
Interferon-γ-R	5′-TGCTTTGCGTTGGACATTCA-3′	
HERV-K *env*-F	5′-CACAACTAAAGAAGCTGACG-3′	[67]
HERV-K *env*-R	5′-CATAGGCCCAGTTGGTATAG-3′	
TNF-α F	5′-AAGCCTGTAGCCCATGTTGTAGC-3′	[40]
TNF-α R	5′-GCCCCTCCACCATGTACTCCTCACC-3′	
GAPDH F	5′-TGAAGGTCGGAGTCAACGGAT-3′	[68]
GAPDH R	5′-TTCTCAGCCTTGACGGTGCCA-3′	

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
