# Peer review of "Activation of Transposable Elements in Immune Cells of Fibromyalgia Patients"

_ijms, 2020, doi:10.3390/ijms21041366_

Round 1

Reviewer 1 Report

This an interesting pilot study investigating if overexpression of HERV’s in PBMC’s from fibromyalgia (FM)patients  can be linked to altered expression of inflammatory  cytokines and t-RNA levels.

The data related to HERV’s is clear in figure 1. The data in fig 2 is Ok but could perhaps be presented better. The data in figure 3 does not appear to correlate with the conclusions in the results.

Figure 2.

In figure 2 the authors show that INF-β and INF-γ are increase in FM patients compared to controls. In the text the authors say patients with increased expression of HERV’s also presented higher levels of INF-B and INF-G). The authors do not show this and these is no direct correlation by linking the two parameters on a plot (i.e one on the x-axis and one on the y)

If data in fig1 and 2 was derived from the same RNA samples the authors could directly make this link by plotting relative expression of the HERV transcripts against the INF-B and IN F gamma transcripts. This would allow the authors to make the statement above possible. If the controls are plotted on the same graphs simple regression analysis will be possible and produce some potentially interesting correlations. At the moment Fig 1 and 2 can not be linked and the conclusions made should not be made

Relating to Figure 2

How does the PBMC values of INF-β and INF-γ compare to levels of actual cytokine levels in the plasma? This would be useful data if available

Figure 3 This is not clear and I wonder if miss labelling has occurred on the figure. In the figure shown there appears to be no difference between levels of tRNA His and tRNA pro for the controls and FM patients although this stated in the text. The figure just shows a difference between t-RNA His and t-RNA pro for patients and controls. The levels between controls and FM patients by looking at the error bars do not look significantly different.

Again a correlation is made with HERV transcipt levels and tRNA levels but this is not directly demonstrated. This could be combined if the data from figures 1 and 3 were combined once the figure is sorted out. Fig 2 and Fig 3 could also be combined to correlate INF-β and INF-γ with tRNA levels.

Author Response

All authors would like to show our gratitude to the reviewer for his/her thorough review of the document and the constructive comments received. We detail the response to each of the comments received in red ink.

This an interesting pilot study investigating if overexpression of HERV’s in PBMC’s from fibromyalgia (FM)patients  can be linked to altered expression of inflammatory  cytokines and t-RNA levels.

The data related to HERV’s is clear in figure 1. The data in fig 2 is Ok but could perhaps be presented better. The data in figure 3 does not appear to correlate with the conclusions in the results.

Figure 2.

In figure 2 the authors show that INF-β and INF-γ are increase in FM patients compared to controls. In the text the authors say patients with increased expression of HERV’s also presented higher levels of INF-B and INF-G). The authors do not show this and these is no direct correlation by linking the two parameters on a plot (i.e one on the x-axis and one on the y)

If data in fig1 and 2 was derived from the same RNA samples the authors could directly make this link by plotting relative expression of the HERV transcripts against the INF-B and IN F gamma transcripts. This would allow the authors to make the statement above possible. If the controls are plotted on the same graphs simple regression analysis will be possible and produce some potentially interesting correlations. At the moment Fig 1 and 2 can not be linked and the conclusions made should not be made

Response: we thank the reviewer for this observation. Since the variables were measured in the exact same set of samples, we subjected data sets to linear regression analysis finding a few interesting, statistically significant, correlations. The results of the analysis have been incorporated as new Figures 3 and 4.

Relating to Figure 2

How does the PBMC values of INF-β and INF-γ compare to levels of actual cytokine levels in the plasma? This would be useful data if available

Response: we agree with the reviewer these are variables to measure in future studies. Unfortunately, we have not included this analysis in the present study.

Figure 3 This is not clear and I wonder if miss labelling has occurred on the figure. In the figure shown there appears to be no difference between levels of tRNA His and tRNA pro for the controls and FM patients although this stated in the text. The figure just shows a difference between t-RNA His and t-RNA pro for patients and controls. The levels between controls and FM patients by looking at the error bars do not look significantly different.

Response: we apologize for the errors included in this figure including an improper selection of data sets for the statistical analysis and an error in the number of samples analyzed (total of 8 samples). The mistakes have now been corrected.

Again a correlation is made with HERV transcipt levels and tRNA levels but this is not directly demonstrated. This could be combined if the data from figures 1 and 3 were combined once the figure is sorted out. Fig 2 and Fig 3 could also be combined to correlate INF-β and INF-γ with tRNA levels.

 Response: no correlations were detected between tRNA and the rest of variables analyzed. This could be due to the low number of samples included in the tRNA analysis and the limitation of the northern blot technique for tRNA and tsRNA detection.

Reviewer 2 Report

This manuscript covers an interesting area related to fibromyalgia, however the presentation of the contents occasionally lacks clarity and thereby limits the understanding for the reader.

1. Results, first section. Details of the demographics of the participants is very limited with respect to the heterogeneity of fibromyalgia. More details need to added that could impact on the outcomes of the study; for example, duration of condition since diagnosis, current/recent medication use.

2. Section 2.1, second paragraph states 'itemized FIQ scores for participating patients are shown...' however only selected items and not the full FIQ items are shown. Please include the full profile of FIQ scores.

3. Results, section 2.3. The text and Figure 2 refers to cytokine (INF and TNFa) levels and cytokine mRNA levels. Clarity is required regarding what the data represents and the text needs revision.

4. Results, section 2.4. Text states 'found reduced levels of tRNA-His and tRNA-Pro in PBMCs of the FM patients' however Figure 3 B shows either raised or similar levels in FM patients relative to HCs. Further in Figure 3B indicates statistical comparisons were made between the HCs measurements and between the FM measurements and not between FM compared to HC values. This figure is confused and needs correcting.

5. The discussion section many aspects of speculation rather than interpretation and conclusions from the data generated. Text needs revision with a focus on the context and value of the data from the study.

5. Results section should not contain references or citation, the context of the work should be explained in the introduction and methods sections. Please revise the text.

6. The text should be written in 3rd person and not include first or second person, such as 'we'. Please revise the text.

Author Response

All authors would like to show our gratitude to the reviewer for his/her thorough review of the document and the constructive comments received. We detail the response to each of the comments received in red ink.

This manuscript covers an interesting area related to fibromyalgia, however the presentation of the contents occasionally lacks clarity and thereby limits the understanding for the reader.

Response: We have extensively edited the text to increase clarity and facilitate reader´s understanding.

Results, first section. Details of the demographics of the participants is very limited with respect to the heterogeneity of fibromyalgia. More details need to added that could impact on the outcomes of the study; for example, duration of condition since diagnosis, current/recent medication use.

Response: Thank you for the comment. We have added details for disease duration of participating patients and medication. The last is included as a new supplementary Table S2.

Section 2.1, second paragraph states 'itemized FIQ scores for participating patients are shown...' however only selected items and not the full FIQ items are shown. Please include the full profile of FIQ scores.

Response: Thank you for noticing. Available itemized scores have been included for the three questionnaires as part of a new supplementary Table S1.

Results, section 2.3. The text and Figure 2 refers to cytokine (INF and TNFa) levels and cytokine mRNA levels. Clarity is required regarding what the data represents and the text needs revision.

Response: We have extensively edited the section title and the text to clarify that the levels of the mentioned molecules were determined by RT-qPCR amplification, and therefore correspond with mRNA levels found in PBMCs.

Results, section 2.4. Text states 'found reduced levels of tRNA-His and tRNA-Pro in PBMCs of the FM patients' however Figure 3 B shows either raised or similar levels in FM patients relative to HCs. Further in Figure 3B indicates statistical comparisons were made between the HCs measurements and between the FM measurements and not between FM compared to HC values. This figure is confused and needs correcting.

Response: We apologize for the errors included in this figure including an improper selection of data sets for the statistical analysis and an error in the number of samples analyzed (total of 8 samples). The mistakes have now been corrected.

The discussion section many aspects of speculation rather than interpretation and conclusions from the data generated. Text needs revision with a focus on the context and value of the data from the study.

Response: We have extensively edited the text to highlight main conclusions from the data obtained in this study. However, the presentation of a model (Figure 6) yet to be validated inherently includes speculation. Discussion also allows for speculation to interpret data.

Results section should not contain references or citation, the context of the work should be explained in the introduction and methods sections. Please revise the text.

Response: We agree with the reviewer that the main context of the work should be included in the introduction. However, we need to respectfully argument that the removal of the short comments introduced in some Results sections may limit reader´s understanding of the reasons behind some of the analysis performed. For example, in section 2.4 the readers would not understand what are the reasons for tRNA analysis if the work by Donovan et al., brief introduction and cite is removed.

In addition we would like to take the reviewer´s attention to the fact that some recent publications on IJMS include references in the Results section. For example:

Gao Q, Li G, Sun H, Xu M, Wang H, Ji J, Wang D, Yuan C, Zhao X. TargetedMutagenesis of the Rice FW 2.2-Like Gene Family Using the CRISPR/Cas9 System Reveals OsFWL4 as a Regulator of Tiller Number and Plant Yield in Rice. Int J Mol Sci. 2020 Jan 26;21(3). pii: E809. doi: 10.3390/ijms21030809.

The text should be written in 3rd person and not include first or second person, such as 'we'. Please revise the text.

Response: Although we acknowledge that scientific narrative is characterized by impersonal voice, we respectfully disagree with the reviewer that the term “we” cannot be used and want to argument the change is more closely related to style rather than pertinence. Nevertheless, we have proceeded to edit the text to remove some of the terms “we” by impersonal sentences. For example: on lines 115 and following  the words “we may be able to observe” has been replaced by the expression “may be observed”; and on lines 150 and following “we measured” has been replaced by “were measured”.

Round 2

Reviewer 1 Report

The paper is now Ok to publish